# Neuro-Urology: Call for Universal, Resource-Independent Guidance

**DOI:** 10.3390/biomedicines11020397

**Published:** 2023-01-29

**Authors:** Glenn T. Werneburg, Blayne Welk, Marcio A. Averbeck, Bertil F. M. Blok, Rizwan Hamid, Michael J. Kennelly, Limin Liao, Stefania Musco, Pawan Vasudeva, Thomas M. Kessler

**Affiliations:** 1Department of Urology, Cleveland Clinic Foundation, Cleveland, OH 44195, USA; 2Department of Surgery and Epidemiology & Biostatistics, Western University, London, ON N6A 3K7, Canada; 3Department of Urology, Moinhos de Vento Hospital, Porto Alegre 90035-000, Brazil; 4Department of Urology, Erasmus Medical Center, 3015 GD Rotterdam, The Netherlands; 5Department of Neuro-Urology, London Spinal Injuries Unit, Royal National Orthopaedic Hospital, Stanmore HA7 4LP, UK; 6Department of Urology, Atrium Health, Carolinas Rehabilitation, Charlotte, NC 28203, USA; 7Department of Urology, China Rehabilitation Research Centre and Capital Medical University, Beijing 100068, China; 8Department of Neuro-Urology, Azienda Ospedaliera Careggi, 50134 Florence, Italy; 9Department of Urology & Renal Transplant, V.M. Medical College and Safdarjang Hospital, New Delhi 110029, India; 10Department of Neuro-Urology, Balgrist University Hospital, University of Zürich, 8008 Zürich, Switzerland

**Keywords:** neurogenic lower urinary tract dysfunction, neurogenic bladder, neurogenic bowel, sexual dysfunction, voiding dysfunction, detrusor overactivity, clinical practice guidelines, urinary tract infection, cystatin C, chronic kidney disease

## Abstract

Neurogenic lower urinary tract dysfunction (NLUTD), the abnormal function of the lower urinary tract in the context of neurological pathology, has been the subject of multiple efforts worldwide for the development of clinical practice guidelines. These guidelines are based on the same body of evidence, and are therefore subject to the same gaps. For example, sexual and bowel dysfunction in the context of NLUTD, optimal renal function assessment in those who are non-ambulatory or with low muscle mass, optimal upper tract surveillance timing, and modification of diagnostic and treatment modalities for low-resource nations and communities are inadequately addressed. In addition, many aspects of the conclusions and final recommendations of the guidelines are similar. This duplicative work represents a large expenditure of time and effort, which we believe could be focused instead on evidence gaps. Here, we call for a global unified approach to create a single, resource-independent, comprehensive guidance on NLUTD, neurogenic sexual, and neurogenic bowel dysfunction. Targeted research addressing the evidence gaps should be called for and pursued. This will allow for focus to shift to filling the gaps in the evidence for future guidelines.

Neuro-urology is a dynamic and rapidly developing subspeciality bridging the fields of neurology and urology (Figure 1) [1,2]. Although neuro-urology lacks the prestige of cancer or heart disease, it is becoming more and more popular, especially considering the increasingly aging population and the high prevalence of neurological diseases. Neurogenic lower urinary tract dysfunction (NLUTD) is the abnormal function of the bladder and/or urethra in the context of a relevant neurological disorder [3]. NLUTD may be the result of a wide range of neurological pathologies, and its manifestations and management are heterogenous. NLUTD is present in up to 75% of those with multiple sclerosis after 10 years, 96% of individuals with spina bifida, 50% of those with Parkinsonian syndrome at disease onset, 83% following stroke, and 95% following spinal cord injury [4]. Due to improvements in the management of such neurological conditions, focus is shifting to not only long-term protection from renal dysfunction due to pathological changes in the lower urinary tract, but also to the optimization of patients’ quality of life and other common NLUTD sequelae such as urinary incontinence and urinary infections. The importance of this has been reflected in the multiple efforts from organizations around the world which have created guidelines on the diagnosis and management of NLUTD. Examples of these societies and medical bodies are shown in Table 1.

Clinical practice guidelines are statements that include recommendations to optimize patient care and are based on systematic evidence reviews and assessments of the benefits and harms of alternative management options [5]. NLUTD guidelines all use an evidence-based approach; however, they also recognize the reality of the limited evidence base for many important questions in NLUTD management (Table 2). Given that the guidelines from each organization are, in large part, based on the same evidence, many of their conclusions and final recommendations are quite similar. For example, all of the major guidelines advise that antimuscarinic medications should be used for storage NLUTD. The major guidelines also agree that in those with medically refractory neurogenic detrusor overactivity, intradetrusor botulinum toxin A injections may be offered. There are many other areas of consensus in the majority of NLUTD guidelines; for example, there is a consistent recommendation against screening or treating asymptomatic bacteriuria, and that intermittent catheterization is the preferable mode of lower urinary tract management when effective and when low-pressure voiding is not possible. For most of these topics, multiple independent assessments of the available evidence have led to similar conclusions (Table 3), both initially and during iterative updates. Although important, this duplicative work represents a large expenditure of time and effort, which we believe could be focused instead on evidence gaps. 

The various NLUTD guidelines are devoid of guidance on several practical and important unaddressed topics germane to the practicing clinician, such as those listed in Table 2. One of the gaps in guidelines is the lack of recommendations for providers in low-resourced countries without access to technology such as urodynamics or supplies such as intermittent catheters. The major guidelines advise that intermittent self-catheterization is preferable to an indwelling catheter. In most well-resourced countries, such as the United States and European countries, intermittent self-catheterization is funded to allow patients to use a new catheter each time they catheterize. In developing countries, patients often do not have access to the necessary catheter supplies, or cannot afford them. In such cases, catheter reuse is required. Multiple types of catheter cleaning techniques have been described, including cleaning with antibacterial soap and water, alcohol sterilization, microwave sterilization, and washing with an aseptic solution such as chlorhexidine 1.5% and cetrimide 15% [6]. A recent study showed that a protocol wherein catheters were treated with detergent and water washing followed by a sodium hypochlorite and sodium chloride solution had bactericidal action on all tested uropathogens [7]. However, clinical studies are lacking, and existing guidelines generally do not provide guidance regarding the cleaning techniques of choice to address situations where catheter reuse is necessary. Similarly, there is scarce guidance on the optimal catheter material, number of uses, and storage conditions if catheter reuse is planned. 

Another major guidance gap is the lack of detailed discussion of the associated sexual or bowel dysfunction which is frequently present with NLUTD. While management of these conditions often has fallen outside the scopes of NLUTD guidelines, the conditions should be considered in parallel with NLUTD because tailored treatments may benefit (or worsen) these other conditions. Data demonstrate that those with worse lower urinary tract symptoms have more severe bowel dysfunction, that stool impaction in the rectum may mechanically impede bladder emptying in neurological patients, and that type of lower urinary tract management correlates with severity of bowel symptoms [8,9,10]. Constipation may be contributory to recurrent urinary tract infection and NLUTD and thus may warrant treatment. Similarly, sexual dysfunction is significantly correlated with urinary incontinence in NLUTD patients [11]. Guidelines currently lack recommendations on penile prosthesis placement at the time of surgical management of neurogenic stress urinary incontinence. 

Existing guidance also generally does not address the optimal process by which a patient with NLUTD is transitioned from pediatric to adult urological clinics. Evidence on this subject is evolving, and some studies suggest that at the time of transition, patients are at risk of declining health [12,13]. Thus, clear guidance is needed regarding a safe and effective transition process but is often outside the scope of the available guidelines, which generally focus on either pediatric or adult care. The guidelines also typically do not address the preferred laboratory assessment of renal function. This may be particularly relevant in individuals with NLUTD who are non-weight-bearing and thus have lower rates of muscle turnover and potentially less accurate estimated glomerular filtration rates (eGFR) when that measure is calculated using serum creatinine (which is muscle-dependent) [14]. Contrarily, cystatin C is a muscle-independent serum marker of renal function and likely represents a superior laboratory assessment of kidney function and marker for renal dosing of therapeutics in this population.

In some cases, specialized guidance documents have been introduced in order to fill in the gaps in the neuro-urology guidance. For example, the Spina Bifida Association provides guidance on the transition of patients with spina bifida from pediatric to adult urological clinics [15]. The international spinal cord injury urodynamics basic data set has been developed [16], and the Infectious Disease Society of America’s catheter-associated UTI guidance addresses infectious manifestations of NLUTD [17]. The nursing and neuro-urological working groups of the Medical Society of Paraplegia have generated guidance on intermittent catheterization in NLUTD [18]. However, these distributed sources are not intuitive to non-specialist physicians and risk being used by only small groups of healthcare providers.

A final area where there is variability in the guidelines is the surveillance protocols for NLUTD. Initially, many guidelines on NLUTD surveillance were written for patients with SCI or spina bifida, where there is a known and significant risk of silent renal deterioration due to lower urinary tract pathologies such as neurogenic detrusor overactivity, detrusor sphincter dyssynergia, low bladder compliance, and vesico-uretero-renal reflux. However, in many cases these guidelines did not represent reality, as few patients receive any degree of surveillance in the real-world setting [19,20,21]. New evidence suggests that even in the first year following SCI, unfavorable urodynamic parameters may be present [22], yet available guidance does not address the timing of early urodynamics assessment in SCI. In addition, these surveillance protocols were not suitable for other NLUTD diseases, such as Parkinsonism, dementia, or stroke. Recent guidelines have recognized that not all NLUTD patients should be surveilled the same way, and they offer a stratified approach to follow-up and routine investigation [23,24]. 

The issues of redundancy, gaps, and variability are not specific to neuro-urology. Other fields have addressed these issues through successfully developing and implementing global consensus guidance. Notably, in 2016 the International Consensus Guidance for Management of Myasthenia Gravis (MG) was released [25]. The guidance sought to “address the lack of uniform, globally accepted standard for the care of people with MG” and was developed using the RAND/UCLA appropriateness criteria by an international panel of 15 experts in the field. It serves as formal guidance for clinicians caring for patients with MG worldwide.

We suggest a unified approach to create a single comprehensive guidance on neuro-urology, including NLUTD, but also neurogenic sexual and bowel dysfunction. Endorsement by the major urological societies would be essential. Guidance for both high-income and low-income countries should be included, and great care must be taken to identify and include guideline panel representatives from diverse geographical locations. Targeted research addressing the evidence gaps should be called for and pursued. The guidance panel should include neuro-urologists and neurologists, but also consultants with expertise in diverse areas including gastroenterology, nutrition, infection, pediatrics, gynecology, and sexual medicine. The guideline should be applicable to all of those who care for neuro-urological patients—urologists, neurologists, primary care providers, specialized nurses, physiatrists, and infectious disease specialists—while recognizing that other specialties may require their own guidance documents to address specific issues within their disciplines. We do not need more summaries of the evidence for treatments that are accepted and have been studied with high-quality randomized controlled trials, nor do we need a weighty textbook that tries to summarize every study in neuro-urology. Well-written, concise, and internationally agreed-upon standards for the management and follow-up of neuro-urological patients will allow focus to shift to filling the gaps in the evidence for future guidelines.

## Figures and Tables

**Figure 1 biomedicines-11-00397-f001:**
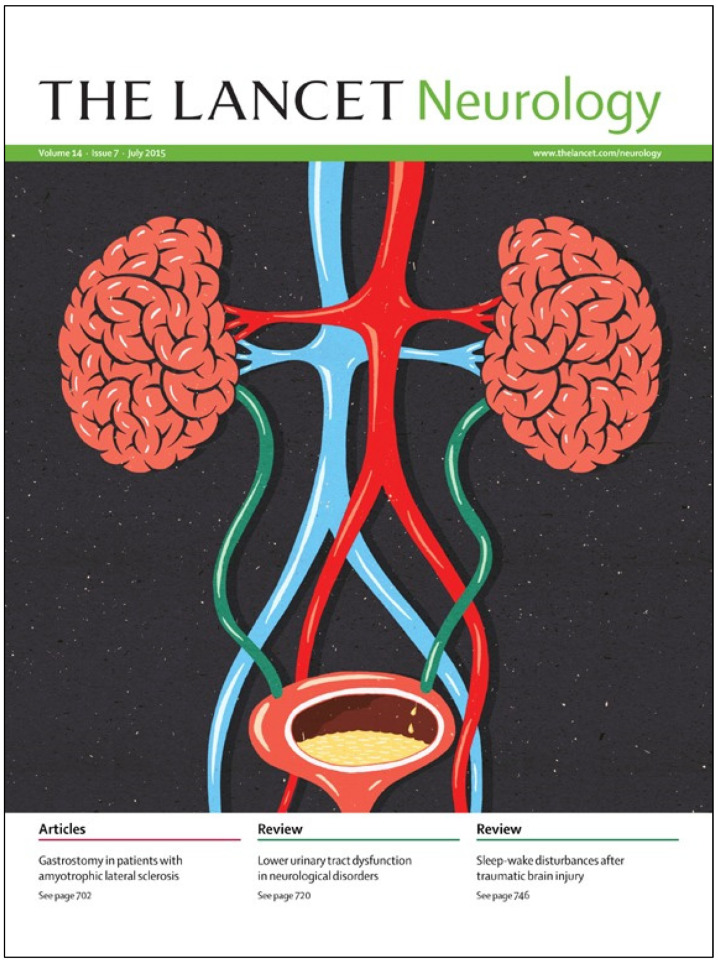
Neuro-urology is a subspeciality bridging the fields of neurology and urology, symbolized by this illustration of the kidneys as brains. Cover picture from *The Lancet Neurology, Volume 14, Issue 7, July 2015* (reproduced with permission).

**Table 1 biomedicines-11-00397-t001:** Organizations with associated neuro-urological guidelines.

Organization	Website (Accessed 19 January 2023)
American Urological Association (AUA)	www.auanet.org
Brazilian Medical Association (AMB)/Brazilian Urological Society (SBU)	www.amb.org.br
Canadian Urological Association (CUA)	www.cua.org
Chinese Urological Association (CUA)	www.cuan.cn
European Association of Urology (EAU)	www.uroweb.org
International Consultation on Incontinence (ICI)	www.ics.org/ici7
International Continence Society (ICS)	www.ics.org
Japanese Urological Association (JUA)	www.urol.or.jp/en
National Institute for Health and Care Excellence (NICE)	www.nice.org.uk
Taiwanese Continence Society (TCS)	www.tcs.org.tw/english/index01.asp

**Table 2 biomedicines-11-00397-t002:** Examples of areas lacking a strong evidence base from which to make neurogenic lower urinary tract dysfunction guidance statements.

Example
Sexual dysfunction evaluation and management
Bowel dysfunction evaluation and management
Diagnostic and/or treatment modification for low-resource settings
Uro-oncological screening (surveillance cystoscopy, cytology, PSA)
Transition from pediatric to adult urological clinic
Reproductive medicine
Optimal renal function assessment
Optimal surveillance timing for lower and upper tract deterioration risk
Geriatric care
Sexual dysfunction evaluation and management

**Table 3 biomedicines-11-00397-t003:** Significant overlap exists in major guidelines regarding the workup and management of neurogenic lower urinary tract dysfunction.

Topic	AUA/SUFU	EAU	NICE	CUA	ICI
History	x	x	x	x	x
Physical Examination	x	x	x	x	x
Risk Stratification	x			x	
Urinalysis	x	x	x	x	x
Post-void residual	x	x	x	x	x
Renal function laboratory assessment	s	x		s	x
Bladder diary	s	x	x	s	s
Renal/bladder ultrasound	s	s	s	s	x
Urodynamics	s	x	s	s	s
IC preferable bladder drainage assistance	x	x		x	x
Antimuscarinic agents for storage	x	x	x	x	x
Beta-3 agonist agents for storage	x				
Alpha-blocker agents for voiding	x	x	x		x
Botulinum toxin	x	x	x	x	x
Sacral neuromodulation	s			s	
Tibial nerve stimulation	s			s	
Urethral bulking agents	s				s
Slings	x	x	x		x
Artificial urinary sphincter	x	x	x		x
Bladder augmentation	x	x	x	x	x
Urinary diversion	x		x		x
Upper tract surveillance protocol	s	s	s	s	s

x: recommended generally in respective guidelines; s: recommended in selected cases in respective guidelines; AUA: American Urological Association; SUFU: Society of Urodynamics, Female Pelvic Medicine & Urogenital Reconstruction; EAU: European Association of Urology; NICE: National Institute for Health and Care Excellence; CUA: Canadian Urological Association; ICI: International Consultation on Incontinence; IC: intermittent catheterization.

## Data Availability

Not applicable.

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
