# Peer review of "Neuro-Urology: Call for Universal, Resource-Independent Guidance"

_biomedicines, 2023, doi:10.3390/biomedicines11020397_

Round 1

Reviewer 1 Report

The authors call for a global unified approach to create a single, resource-independent, comprehensive guidance on NLUTD, neurogenic sexual and neurogenic bowel dysfunction. This suggestion is widely supported and could be extended to others urologic fields.

The main goals are 1) to avoid repetitions; 2) to include recommendations for low-resourced countries; 3) to fill the gaps on some unaddressed topics; 4) to create comprehensive guidance on neuro-urology, including NLUTD, but also neurogenic sexual and bowel dysfunction

I strongly support this call and I hope to read it in the short future.

Author Response

We thank the reviewer for the very positive feedback.

Reviewer 2 Report

The manuscript by Werneburg and colleagues is focused on neuro-urology guidelines and how they can be improved.They defined neuro-urology as being an intersection between neurology and urology, rapidly growing to become a true medical subspeciality. Neuro-urology deals with dysfunciton of the lower urinary tract associated with neurological impairment, inlcuding multiple sclerosis, spinal cord injury and Parkinson's disease. While classically neuro-urological interventions were aimed to protect the upper urinary tract, current approaches are more focused on improving the quality of life in the long term. This has been recognized by several national organizations that have produced guidelines. While there is agreement in several points, not all guideline produce the same recommendations. This likely reflects the socio-economic context in which guidelines are produced. Considering the lack of strong recommendations in some areas and lack of consensus in some points, the authors call for the production of improved guidelines that should be more consensual and include previously unregarded areas.

I agree with the auhors and my only big suggestion is that the authors should stress that imrpoved guidelines should be produced by a group of healthcare professionals that is not restricted to urologists and neurologists. For example, if recommendations for bowel management are to be included, then gastroenterologists and nutritionists should be consulted. Paediatricians, gynaecologist, specialists in sexual medicine... these are all specialisties that may produce helpful contributions.  Importantly, care should be taken to include specialists from different countries, that have different legislations, health care systems and cultural backgrounds. This would allow adjustment for different socioeconomic contextes.

A minor point: in the second page, the transition between the 1st paragraph, ending in line 107, to the 2nd paragraph, beginning in line 111, seems a bit blunt. While it is important to mention the transition from paedeatric to adult clinical care, the 2nd paragraph seems a bit off. Maybe there were parts that were eliminated during editing?

Author Response

We appreciate the reviewer's comments and have modified the text to address all the points. Specifically, in the final paragraph wherein we provide detail on the scope and development of the guidelines, we have now included points regarding the geographical diversity of the panel, as well as representatives with expertise in the areas suggested by the reviewer. In addition, we have modified the wording in lines 111-113 to improve this transition.